# Improved Power Normalized Cepstrum Coefficient Based on Wavelet Packet Decomposition for Trunk Borer Detection in Harsh Acoustic Environment

**Huanyu Zhou** [1,†] , **Ziqi He** [1,†] , **Liping Sun** [1,†,*] , **Dongyan Zhang** [1,*] , **Hongwei Zhou** [1,†] **and Xiaodong Li** [2]

1   College of Mechanical and Electrical Engineering, Northeast Forestry University, Harbin 150040, China; zhy17801034727@163.com (H.Z.); ziqihe123456@163.com (Z.H.); easyid@163.com (H.Z.)
2   General Station of Forest and Grasland Pest Managenment, National Forestry and Grassland Administration, Shenyang 110034, China; zhy17801034727@gmail.com
*   Correspondence: SunLiping123123@163.com (L.S.); nefuzdhzdy@nefu.edu.cn (D.Z.); Tel.: +86-1584-638-8678 (L.S.); +86-1513-456-0311 (D.Z.)
†   These authors contributed equally to this work.

**Abstract:** The sound-detection method of trunk borer is a very promising method in the field of forestry prevention and control of trunk borers. However, the detection accuracy of commonly used algorithms often decreases sharply in the case of noise reverberation interference. In practical applications, the sound monitoring of trunk borers often takes place in a harsh acoustic environment. To solve this problem, we intend to introduce methods which are effective in other related acoustic fields. Unfortunately, most of the methods are not suitable for acoustic detection of trunk borers and perform extremely poorly. After trying various methods, we found that Power-Normalized Cepstral Coefficients (PNCC) performed well in some cases, while it did not in others. This is due to the difference between speech and trunk borer sound. Therefore, an improved anti-noise PNCC based on wavelet package is proposed. The dmey wavlet system always obtains the best performance. We collected the audio of the following five dry borer pests for testing. They are red palm weevil, mountain pine beetle, red necked longicorn, Asian longhorn beetle and citrus longhorn beetle. In the experimental part, we used genetic algorithm-support vector machine (GA-SVM) as a classifier to compare Mel Cepstral Coefficients (MFCC), which are the most common methods in the field of audio detection of trunk borer, PNCC and improved PNCC in a variety of noise environments. The results showed that, compared with other methods, the newly proposed method can often achieve better results. The above experiments take the audio clips made of clear pest sound mixed noise. In order to further verify the effectiveness of the method, we designed another experiment with a harsh outdoor acoustic environment. We found that the proposed method achieved 88% accuracy and the traditional PNCC achieved 78% accuracy. However, the Mel cepstrum coefficient completely lost its ability to distinguish. In sum, the proposed PNCC based on wavelet packet decomposition can be used as a detection method for trunk borer in the harsh acoustic environment. This method has many advantages, including simple extraction and strong robustness to noise. Combined with cheap audio acquisition equipment, this method can effectively improve the early warning ability of forestry borer pests.

**Keywords:** power normalized cepstrum coefficient; trunk borer; sound detection; wavelet packet transform

## 1. Introduction

Trunk borer mainly refers to all kinds of longicorn beetles, gibberries, weevil beetles, bark beetles, Lepidoptera and wood beetles. Among the forest diseases caused by pests, trunk borer has become the most difficult to control in China because of its hidden living habits and the slow performance of damaged trees. In recent years, trunk borer has caused serious harm to the Chinese forest industry. The occurrence area in 2019 alone was

about 22 million acres [1] . Among truck borers, in most areas south of the Yangtze River, Monochamus alternatus is rampant. The forest areas of southern and eastern Heilongjiang, Jilin and eastern Liaoning are infected by Chilo suppressalis. In some areas of Southwest China, Dendroctonus can cause disaster. The eight-toothed bark beetle destroyed trees in southeastern Inner Mongolia and eastern Tibet. The Huashan pine bark beetle lives in southern Gansu. In the eastern part of Inner Mongolia, the Hexi region of Gansu Province and Guanzhong Plain of Shanxi Province, the harm of stem borers such as Anoplophora glabripennis continues to worsen [1].

Thus, it can be seen that the problem of forestry trunk borer control in China urgently has to be settled. However, the traditional detection method of forest trunk borers is not only time-consuming and laborious, but also inefficient. Sound detection technology has a good development prospect in improving detection accuracy, with less time and low cost. Scholars from all over the world have made some achievements in using sound detection technology to detect different kinds of pests. Mankin et al. [2–4] summarized the acoustic characteristics of honey dragon larvae in sugarcane in Australia, rhinoceros horns (Coleoptera: Chrysomelidae: termites) and termites horns (Isoptera: termites) in palm trees in Guam, and the branches of Monochamus (Coleoptera: Cerambycidae) in the second Sea America (Laurus: Laurales: Lauraceae). Hetzroni et al., 2016, used a piezoelectric sensor to capture and diagnose the vibration signals of red palm weevil larvae living mainly in palm trees such as jujube and canaries [5]. Lemos Escola et al., 2020, designed a sound detection method based on wavelet packet transform and support vector machine to monitor the main pests affecting coffee production in South American countries [6]. According to the acoustic characteristics of stem borer larvae, Sutin et al., 2019, proposed an automatic acoustic detection algorithm for wood borer larvae [7]. Bilski et al., 2017, used acoustic emission technology to realize the early monitoring of borer larvae in wooden furniture and buildings [8]. Pan et al., 2015, summarized the different time domain and frequency domain distribution characteristics of double-hook Heteroptera beetle larvae [9]. Siriwardena et al., 2010, used a portable and efficient recording device to record and analyzed the sound of the red palm weevil larvae, which mainly live in the coconut tree [10]. Eliopoulos et al., 2015, used sound emission technology to detect adult beetles living in wheat [11]. Zhao et al. [12] explored the sound detection technology that can be used for the (Semanotusbifasciatus) larvae of longicorn beetles. At present, the research on sound detection of trunk borer pests basically depends on the laboratory acoustic environment, without background noise interference. However, in real application, the effects of noise and all kinds of interference on the system must be considered. In this study, five trunk borer pests, namely, red palm weevil (Rhynchophorus ferrugineus), mountain pine beetle, red necked longicorn (Aromia bungii), Asian longhorn beetle (Anoplophora glabripennis) and citrus longhorn beetle (Anoplophora chinensis), were used as experimental objects. As shown in the Figure 1, the pictures indexed by the tags are photos of thesr five trunk borers. The trunk borers we describe in the following chapters are these five kinds of pests.

In the next section, we elaborate on the state-of-the-art related field methods that may potentially be applied to this problem and explore their feasibility. They include research results from signal processing, sound event classification, sound scene classification, speech recognition and other related fields.

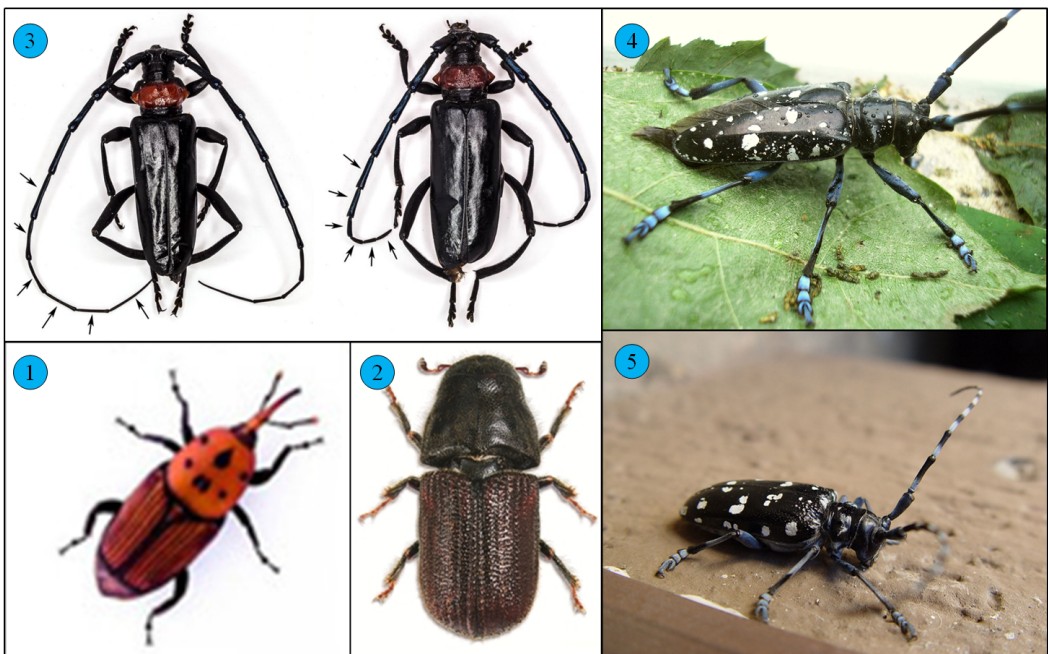

**Figure 1.** Photos of typical trunk borers: (**1**) red palm weevil (Rhynchophorus ferrugineus) [13], (**2**) mountain pine beetle, (**3**) red necked longicorn (Aromia bungii) [14], (**4**) Asian longhorn beetle (Anoplophora glabripennis) and (**5**) citrus longhorn beetle (Anoplophora chinensis) in turn.

## 2. Feasibility Analysis of Related Work

In recent decades, the performance of sound detection system in a good acoustic environment has been significantly improved. However, most sound detection systems are still sensitive to the nature of their input data's acoustic environment, and their performance deteriorates sharply in the presence of degraded sources such as additive noise, linear channel distortion and reverberation. In recent years, experts and scholars have introduced dozens of algorithms to solve these problems, and some methods have achieved some results in their field. However, the audio of different things often has different acoustic characteristics. Generally speaking, specific methods only work on specific things.

Generally speaking, the features used in the field of sound detection are divided into the following categories:

- Basic features;
- Time domain features;
- Frequency domain features;
- Time–frequency domain features;
- Other transform domain features;

In recent years, neural networks have been used to automatically screen out sound features.

Some basic features, such as loudness, tone, duration and timbre, are not applicable in this study. Time domain features like short-term energy, short-term average amplitude, short-term average zero-crossing rate and many other features are highly sensitive to noise. These are also not suitable for this study. Frequency domain features include sound signal short-time spectrum, short-time spectrum critical band vector features, etc. Such features are now often used as a process in more advanced features, rather than used alone.

Time-frequency domain features are commonly used in Acoustic Event Classification (AEC) and Auditory Scene Analysis (ASA). They included a spectrogram, spectral map, cochlear map and image features of power distribution of offspring (SPD) [15]. In recent years, the robust texture features [16] based on Gamma filter logarithmic spectrum, and Fusion Fisher Vector features (FFV) [17] are also two-dimensional time-frequency image features. From a perceptual point of view, this kind of method seems to be helpful to this

study. In fact, we also tried some more advanced methods, such as SPD and FFV. However, their detection effect is not good. Their performance is even weaker than the traditional MFCC method in the field of trunk-borer audio detection. This is because the vibration time caused by trunk borers is shorter (as shown in Figure 2), but the duration of the sound event signal is often longer. This kind of advanced method usually abandons frame-based recognition and processes the sound event on the basis of two-dimensional time-frequency images. This is effective for the classification of sound events, such as applause, chair moving, footsteps, phone ringing, or a door slamming in the meeting room environment. However, it is bad for this study. In this study, noise reverberation obscures the sound information of trunk borer to a great extent (as shown in Figure 3), and it is difficult to capture useful information in a long time-range.

Other transform domain features include the wavelet domain of the wavelet transform and the cepstrum domain of multiple cepstrum coefficients. In addition, the complex cepstrum domain and the complex signal domain represented by empirical mode decomposition (EMD) and variational mode decomposition (VMD) [18] are also included. In the research, we also tried to use advanced digital signal processing methods such as VMD to reduce noise. The results show that this method is effective in the case of weak noise amplitude (here, we give the VMD parameters: the number of modal components is 8 and the penalty factor is 2000). However, in the harsh acoustic environment, the method is completely ineffective, and the frequency centers of each modal component are concentrated on the main frequency components of the noise.

Some scholars use neural networks to extract sound features, such as auto-encoder [19,20], convolutional neural network (CNN) [21,22], recurrent neural network (RNN) [23,24], etc. However, these methods require larger training data to learn, and it is difficult to obtain significant features from the learned neural network model. Therefore, such methods are not considered in this study.

At present, the Mel Cepstrum Coefficient (MFCC), wavelet transform, wavelet packet transform and LPCC linear cepstrum coefficient are widely used in trunk borer sound detection. However, these methods are less robust to noise, that is, when the acoustic environment includes interference such as additive noise, channel distortion and reverberation, the recognition accuracy of the algorithm is significantly reduced. Considering that the duration of the vocal behavior of a trunk borer is about 15 ms (as described in Section 3.1), which is about the same as the duration of a frame signal, a short-time frame should be preferred. Some state-of-the-art cepstrum coefficients are usually based on a short time frame. In recent years, the most commonly used cepstrum coefficients are PNCC and constant Q cepstral coefficients (CQCC). Both of them are new methods in the field of speech processing. CQCC focuses on speaker recognition, while Power Normalized Cepstrum Coefficient (PNCC) focuses on speech recognition in noisy environments.Therefore, PNCC is more suitable for this study.

Power Normalized Cepstrum Coefficient (PNCC) [25] is inspired by MFCC and can be regarded as an improved method based on MFCC. Considering the achievements of generalized MFCC in this area, we firmly believe that this method will be suitable for this study. The results in Section 5 show that its performance is better than that of MFCC in most noise reverberation environments. However, we believe that the detection ability of this method should not be limited to this, because all the parameters and processing processes of this method are completely designed around the characteristics of human speech, not a trunk borer. Therefore, we improve this in Section 3.

## 3. PNCC Method Based on Wavelet Packet Transform

PNCC, like MFCC, comes from the field of automatic speech recognition. In view of its excellent ability to supress background noise, it has been widely used in many fields. However, at present, no researchers have tried to apply it to the field of audio detection of a trunk borer. In Section 2, we explained the reasons for choosing this method and the necessity of improvement. In this section, after the audio analysis of several common trunk

borers, the PNCC algorithm is improved according to the time-frequency characteristics of trunk borer audio, which makes it more suitable for the detection of trunk borers in a harsh acoustic environment.

### 3.1. Raw Data Analysis and Preprocessing

In order to improve the PNCC method, we first analyze the audio signals collected in the acoustic environment of the laboratory (as shown in Figure 2).

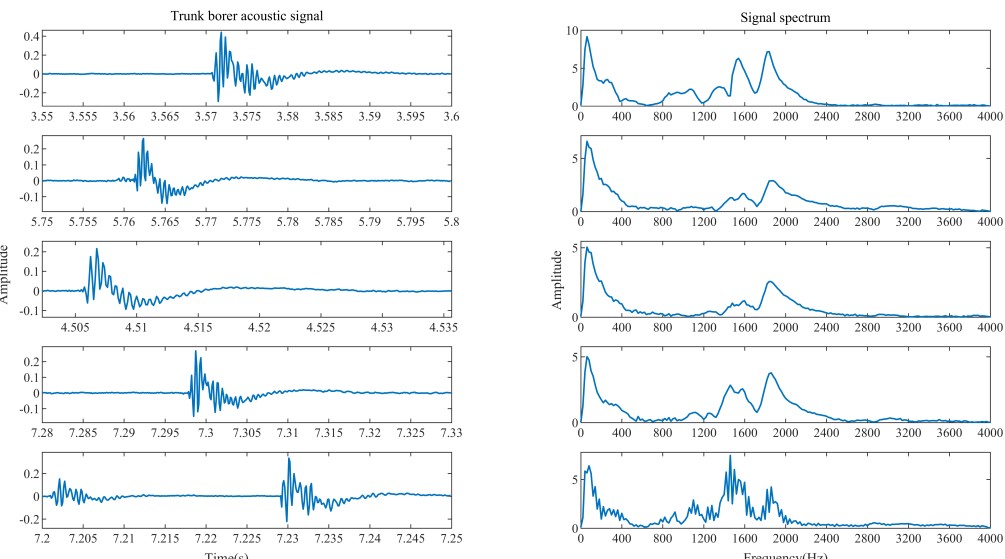

**Figure 2.** This picture shows the audio clip and its spectrum, recorded in the laboratory acoustic environment with a sampling frequency of 8000 Hz. In this study, this is considered to be the ideal audio data of trunk borers, without any interference.

According to the communication standard of the international digital telephone, we thought that the frequency range of human voice signal is mainly concentrated between 300 and 3400 Hz. Many parameters of the original PNCC algorithm are determined according to this frequency range. In this study, through Fourier analysis, it is found that the audio range of trunk borer mainly consists of two bands: 0–400 Hz and 1400–2300 Hz. The peak appears at about 1500 Hz, 1850 Hz and 70 Hz, as shown in Figure 2. Therefore, in the following chapters, we will improve the PNCC according to the main frequency range of stem borer.

It can be observed that there are some differences in frequency distribution among different data in Figure 2. Some audio data include the frequency components of the 1000 and 1300 Hz bands, while others do not. The main reasons for this difference are the species, age and behavior of trunk borers. In addition, we can also observe, from Figure 2, that the vibration duration of the trunk borer signal is about 15 ms.

Generally speaking, when using sound detection technology to monitor trunk borers, the recorded audio data are carried out in a harsh acoustic environment. Figure 3 shows an audio signal in a harsh acoustic environment, which comes from the side of the road. The audio mainly includes the sound of conversation, the roar of cars and the faint sound of trunk beetles eating tree trunks. By observing the frequency spectrum of this signal, we can observe that the audio information of trunk borers is completely obscured by noise and reverberation. Thus, it can be seen that the audio of trunk borer is not easy to observe over a long period of time. In order to obtain better classification results, we pre-filter all the audio data collected in a bad acoustic environment to remove the frequency components which are larger than 2400 Hz or between 400 and 800 Hz as much as possible.

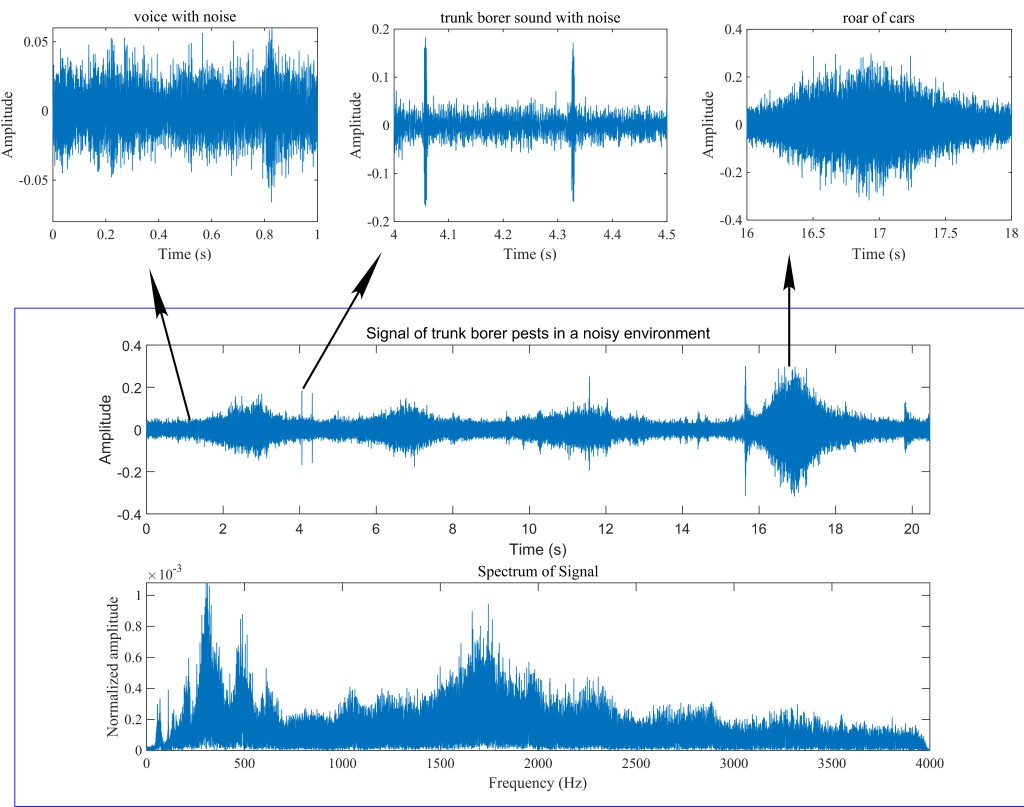

**Figure 3.** This picture shows the audio signal and its frequency spectrum of trunk borers in harsh acoustic environment.

### 3.2. Improved PNCC Method Based on Wavlet Package

The specific calculation process of the original PNCC algorithm can be divided into the following steps: audio signal front-end processing, temporal integration for environmental analysis based on asymmetric noise suppression, mean power normalization and nonlinear processing.

In this study, the first part of the original PNCC algorithm is improved. The overall framework of the algorithm is as follows (Figure 4).

In the audio front-end processing of the original PNCC method, it is often necessary to pre-emphasize the sampled speech signal in the form of $H(z) = 1 - 0.97z^{-1}$. Here, we can understand this as a high-pass filter. Specifically, this is to remove the effect of lip radiation and increase the high-frequency resolution of speech. However, in this study, the audio data frequency distribution of a trunk borer is mainly concentrated in the vicinity of 70, 1500 and 1850 Hz. There is almost no audio component of trunk borers in 400 ~800Hz and a frequency band higher than 2400 Hz. Therefore, if the pre-emphasis processing method of speech signal is used, the audio component of trunk borer around 70 Hz will be ignored and the high-frequency noise between 2400 and 4000 Hz will be amplified. Another reason for this is that there is no lip radiation effect in this study, so we remove the pre-emphasis part of the original algorithm.

After pre-weighting, the original PNCC algorithm uses short-time Fourier transform and a 40-channel Gammatone filter with equivalent rectangular bandwidth center frequency between 200 and 8000 Hz, and the output results are processed in a short- and medium-time, respectively.

In this study, we use wavelet packet transform to replace the short-time Fourier transform in the original method. Compared with short-time Fourier transform, wavelet packet transform develops the localization advantage of short-time Fourier transform and overcomes the disadvantage of constant window function, so it is an effective method for non-stationary signal analysis and feature extraction. The duration of the audio data used

in the study is about 20 s, while the audio signal duration of trunk borer is about 15–20 ms (as shown in Figure 2). From Section 3.1, we know that the trunk borer audio is not easy to observe over a long period of time, so we retain the framing processing and make the frame length close to the duration of the trunk borer audio as possible. Then, the wavelet packet coefficients are obtained by using the three-layer wavelet packet transform in the time-frequency domain, and the low-frequency and high-frequency parts of the spectrum are spliced in the order from low to high. Next, according to the audio frequency range of trunk borers, we adjust the center frequency range of the equivalent rectangular bandwidth of Gammontone filter bank to 50~4000 Hz. Considering that the energy of speech and signals of trunk borers are more concentrated than noise and reverberation signals, we retain the original PNCC method in temporal integration for environmental analysis based on asymmetric noise suppression, mean power normalization and nonlinear processing. The details are as follows.

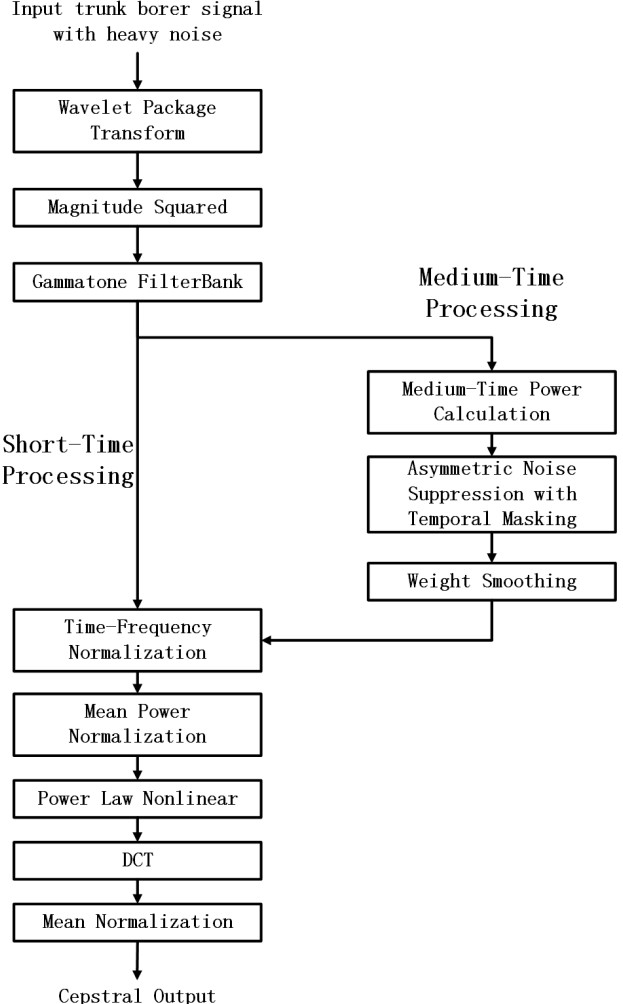

**Figure 4.** Proposed method flowchart.

First of all, the short-term power of PNCC is calculated according to the spectrum. The short-term power of the PNCC feature is defined as follows

$$P[m,l] = \sum_{k=0}^{(K/2)-1} \left| X\left[m, e^{j\omega_k}\right] H_l\left(e^{j\omega_k}\right) \right|^2 \tag{1}$$

where $K$ represents the length of STFT transform, $m$ and $l$ represent input data frame number index and gammontone filter channel index. $P[m,l]$, $X[m,e^{j\omega_k}]$, and $H_l(e^{j\omega_k})$ represent the

short-term power, the frequency response of the input data and the frequency response of the Gammontone filter of the index, respectively.

In PNCC, the background noise processing of audio signal based on mid-time frame analysis. The mid-time power is calculated according to the following Equation

$$\tilde{Q}[m,l] = \frac{1}{2M+1} \sum_{m'=m-M}^{m+M} P[m',l] \tag{2}$$

where $\tilde{Q}[m,l]$ represents mid-time power. In this study, the time integration factor $M$ is 2, because it can better suppress background noise such as white noise.

The asymmetric nonlinear noise suppression filter includes time mask, asymmetric, low-pass filter and other modules. Its complete characteristics can be described by the following Equation

$$\tilde{Q}_{\text{out}}[m,l] = \begin{cases} \lambda_a \tilde{Q}_{\text{out}}[m-1,l] + (1-\lambda_a)\tilde{Q}_{\text{in}}[m,l] \\ \qquad\qquad \text{if} \qquad\qquad \tilde{Q}_{\text{in}}[m,l] \geq \tilde{Q}_{\text{out}}[m-1,l] \\ \lambda_b \tilde{Q}_{\text{out}}[m-1,l] + (1-\lambda_b)\tilde{Q}_{\text{in}}[m,l] \\ \qquad\qquad \text{if} \qquad\qquad \tilde{Q}_{\text{in}}[m,l] < \tilde{Q}_{\text{out}}[m-1,l] \end{cases} \tag{3}$$

where $\tilde{Q}_{\text{in}}[m,l]$ and $\tilde{Q}_{\text{out}}[m,l]$ represent input and output, respectively. Specifically, mid-time power is used as input for filtering. $\lambda_a$ and $\lambda_b$ are constants between 0 and 1. The selection of $\lambda_a$ and $\lambda_b$ values affects the recognition accuracy to some extent. In this study, the values are 0.999 and 0.5. These two values are selected to maximize the recognition accuracy of undisturbed audio and the performance of this method in the case of noise.

Concretely, the lower envelope of the average noise power $\tilde{Q}_{le}[m,l]$ is calculated firstly by

$$\tilde{Q}_{le}[m,l] = \mathcal{AF}_{0.999,0.5}[\tilde{Q}[m,l]] \tag{4}$$

where $\tilde{Q}_{le}[m,l]$ is processed by nonlinear low-pass filter$\tilde{Q}[m,l]$, and its initial value $\tilde{Q}_{le}[0,l]$ is $0.9\tilde{Q}[m,l]$. $\tilde{Q}_{le}[m,l]$generates the rectified output $\tilde{Q}_0[m,l]$ via an ideal half-wave linear rectifier, and then calculates the lower envelope of $\tilde{Q}_0[m,l]$ again

$$\tilde{Q}_f[m,l] = \mathcal{AF}_{0.999,0.5}[\tilde{Q}_0[m,l]] \tag{5}$$

Then, take the larger value in $\tilde{Q}_f[m,l]$ and $\tilde{Q}_{tm}[m,l]$ based on $\tilde{Q}_0[m,l]$ to obtain $\tilde{Q}_1[m,l]$

$$\tilde{Q}_1[m,l] = \max\left(\tilde{Q}_{tm}[m,l], \tilde{Q}_f[m,l]\right) \tag{6}$$

Finally, the final output is determined according to Equation (7)

$$\begin{aligned} \tilde{R}[m,l] &= \tilde{Q}_1[m,l] \text{ if } \tilde{Q}[m,l] \geq c\tilde{Q}_{le}[m,l] \\ \tilde{R}[m,l] &= \tilde{Q}_f[m,l] \text{ if } \tilde{Q}[m,l] < c\tilde{Q}_{le}[m,l] \end{aligned} \tag{7}$$

where $c$ is a fixed constant with a value of 2. It is considered that if the value is less than several times its lower envelope, then it is not a large enough incentive, and vice versa.

Time masking processing is based on the fact that the human auditory system pays more attention to the beginning of the input power. To put this simply, the moving peak of each frequency channel is obtained, and the instantaneous power is suppressed when the instantaneous power is lower than the envelope. Specifically, the following Equation is used to obtain the peak power of each frequency channel

$$\tilde{Q}_p[m,l] = \max\left(\lambda_t \tilde{Q}_p[m-1,l], \tilde{Q}_0[m,l]\right) \tag{8}$$

where $\lambda_t$ is the forgetting factor to obtain peak power. The time mask for audio is implemented using the following Equation

$$\tilde{Q}_{tm}[m,l] = \left\{ \begin{array}{ll} \tilde{Q}_0[m,l], & \tilde{Q}_0[m,l] \geq \lambda_t \tilde{Q}_p[m-1,l] \\ \mu_t \tilde{Q}_p[m-1,l], & \tilde{Q}_0[m,l] < \lambda_t \tilde{Q}_p[m-1,l] \end{array} \right. \tag{9}$$

The best value of $\lambda_t$ and $\mu_t$ is 0.85 and 0.2. In this case, the recognition accuracy is the highest, and the migration attenuation duration calculated from this is about 100 ms, which is consistent with the human auditory system.

After the above processing, it is necessary to smooth the different frequency channels of the Gammatone filter, that is, power-spectrum-weighted smoothing. The above is the temporal integration for environmental analysis of audio signal based on asymmetric noise suppression.

Then, the mean power normalization of PNCC is realized in the form of difference equation. This is because the subsequent power law nonlinearity will cause the processed response to be affected by the changes in absolute power. The mean power normalization process will reduce the potential effect of amplitude scaling on PNCC.

The power law curve of sound pressure index 1/15 fits well with the physiological data, maximizing the recognition accuracy in the presence of noise. Therefore, the power law nonlinear adopts the power law curve with the sound pressure index 1/15. After the output results processed by discrete cosine transform and mean normalization, the PNCC coefficients are obtained.

### 3.3. Wavelet Packet Transform

As described in the previous paragraph, the original PNCC method uses short-time Fourier transform as the front-end processing of audio signal. The audio signal of the trunk borer in a severe noise environment is a typical non-stationary signal. At present, typical non-stationary signal processing methods include short-time Fourier transform (FFT), wavelet transform (WT), empirical wavelet transform (EWT), wavelet packet transform (WPT), empirical mode decomposition (EMD), and variational mode decomposition (VMD). Based on Hilbert transform, EMD and VMD transform the original signal into a complex analytical signal domain for analysis and processing. Short-time Fourier transform, wavelet transform and wavelet packet transform carry out signal processing in the time-frequency domain. Therefore, using wavelet packet transform to improve the original PNCC method can retain the noise processing process of the original PNCC method as completely as possible. At the same time, considering that the wavelet packet transform has a higher high-frequency resolution, it can better listen to the information of the high-frequency components of trunk borer. In fact, we think that, in theory, we can use the VMD or EMD method to replace the preprocessing process of the original PNCC method. However, the subsequent processing process needs to be changed accordingly, and a complete theoretical derivation and practical effect verification are needed at the same time. Considering that the work is too large, and less related to the theme of this study, and the possibility of failure is high, we abandoned this scheme. However, this would be a good direction for future research.

WPT is a universalization of discrete wavelet transform (DWT), and the necessary frequency resolution is able to be realized [26]. Wavelet packet transform achieves more high-frequency resolution at the cost of abandoning time resolution. Figure 5 shows an example of the WPT decomposition tree structure (three levels) of a set of clear audio signals of trunk borers $f(t)$.

The wavelet packet tree is regarded as a filter bank, which is generated by two sets of orthogonal wavelet base filter coefficients

$$\varphi^{2i} = \sqrt{2} \sum_k h(k) \varphi^i(2t-k) \tag{10}$$

$$\varphi^{2i+1} = \sqrt{2} \sum_k g(k) \varphi^i (2t - k) \tag{11}$$

where $g(k)$ and $h(k)$, denoted as group-conjugated orthogonal filters, are quadrature filters related to the mother wavelet function and scaling function, respectively. $\varphi$ stands for wavelet. Each element in the wavelet tree is considered to be a separate filter. The overall response of the filter bank can be described by the following Equation

$$\varphi^i_{j,k}(t) = 2^{j/2} \varphi^i \left( 2^j t - k \right) \tag{12}$$

where $j, i$ and $k$ represent scale, modulation and translation parameters, respectively. WP coefficients of the time domain $f(t)$ are calculated using the following Equation

$$C^i_{j,k}(t) = \int_{-\infty}^{\infty} f(t) \varphi^i_{j,k}(t) dt \tag{13}$$

where $C^i_{j,k}$ wavelet packet coefficient and meets the orthogonality condition:

$$\varphi^m_{j,k}(t) \varphi^n_{j,k}(t) = 0, \quad (m \neq n) \tag{14}$$

The WP component of a signal at the specific node of the decomposition tree is

$$f^i_j(t) = \sum_{k=-\infty}^{\infty} c^i_{j,k}(t) \varphi^i_{j,k}(t) \tag{15}$$

After the decomposition of the $j$th level, the raw signal is built up via the summation of $2^j$ elements, shown as

$$f(t) = \sum_{i=1}^{2^j} f^i_j(t) \tag{16}$$

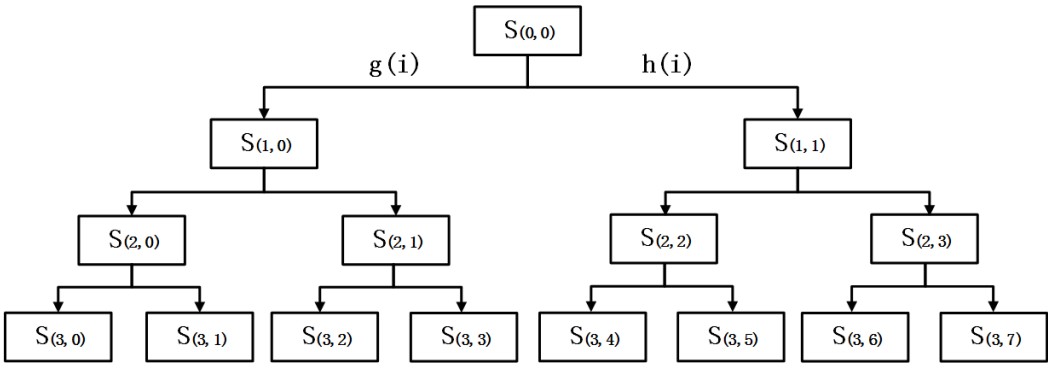

**Figure 5.** Three-level WPT tree structure.

### 3.4. Wavelet Packet Basis Function Selection

Common wavelet basis functions include the Haar wavelet system, Daubechies (dbN) wavelet system, bioorthogonal (biorNr.Nd) wavelet system, Symlets (symN) wavelet system and dmey wavelet system. Generally speaking, the Haar wavelet system is often used in theoretical research, Daubechies wavelet system is often used for signal decomposition and reconstruction, as a filter, and the biorthogonal wavelet system is commonly used in the field of image processing. In order to determine the most suitable wavelet system for the audio data of dry-boring pests, a set of experiments was carried out. In this experiment, we used different wavelet systems to decompose the experimental signals and drew the frequency spectrum before and after the decomposition, as shown in Figure 6.

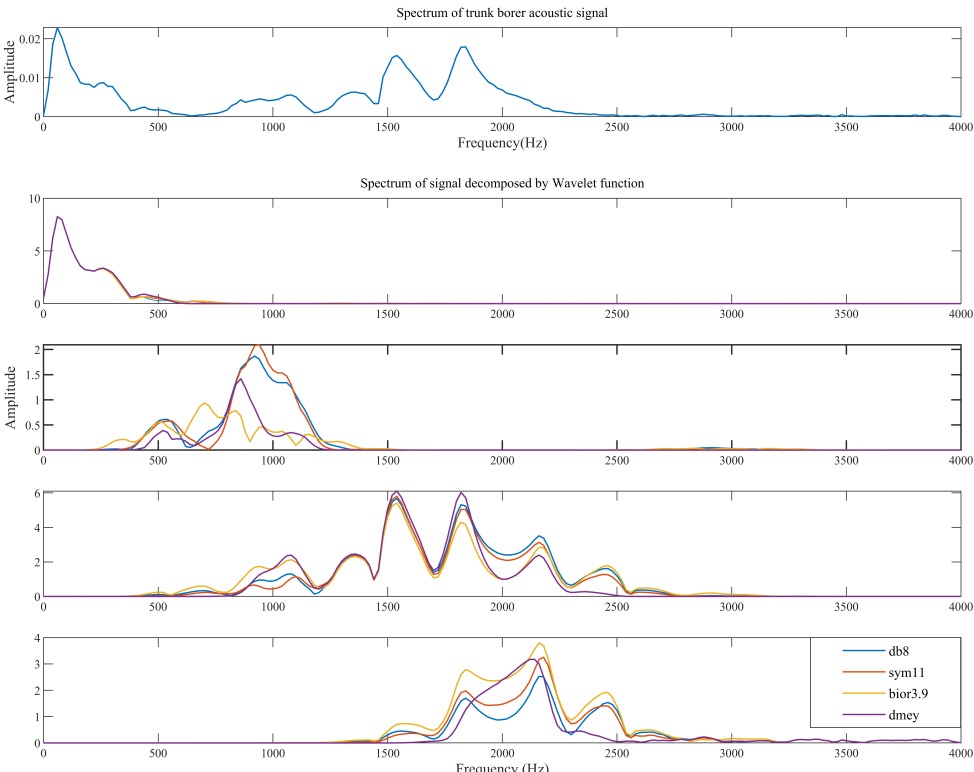

**Figure 6.** This picture expresses the frequency components of the audio signal of clear trunk borers, decomposed by different wavelet systems.

Through observation, it is found that the third and fourth components of db8, sym11, and bio3.9 wavelets have frequency components centered at 2400 Hz. However, the amplitude of the original audio frequency spectrum is about 0 when it is greater than 2300 Hz, which indicates that there are false frequency components. In the 500–1000Hz frequency band, the bior3.9 wavelet system is more chaotic than other wavelet systems. In fact, we tried multiple sets of clear trunk boring pest audio signals, and the results showed similar properties. Therefore, we believe that, in this study, the dmey wavelet system performs best, followed by the sym11 and db8 wavelet systems, and finally the bior3.9 wavelet system. Therefore, the dmey wavelet system is selected as the wavelet packet basis function .

## 4. Sound Recognition of Trunk Borer Using SVM

The reasons for using SVM as the classification model in this study are as follows:

- First of all, although deep neural networks and other related methods are preferred in most harsh acoustic environments, the artificial selection of features with strong robustness to noise, combined with the SVM classification model, is even better than the recognition results of the deep neural network model in some fields of sound recognition [27,28];
- Secondly, the object discussed in this paper is the sound detection of trunk borers in a noisy environment. Compared with the deep neural network, the artificial selection feature combined with machine learning classification method requires fewer data, and the model is relatively easy to train. It is easier to verify the feasibility of the method;
- Thirdly, at present, the main machine learning methods commonly used for audio signal classification are GMM, HMM and SVM. The typical application of support vector machine is to solve the problem of binary classification, that is, to judge whether the test sample belongs to a positive class or negative class. The purpose of this study

is to identify the signals of trunk borer under an adverse acoustic background. In essence, it is a two-classification problem. In addition, the improved PNCC feature is a high-dimensional feature. Therefore, support vector machine is more suitable. Furthermore, SVM's strong generalization ability makes it more suitable for solving problems in practical applications;

- Finally, in this study, when comparing the accuracy of MFCC and PNCC in the sound recognition of trunk borer under the condition of noise, background sound, channel distortion, reverberation interference, or non-synchronous training test environment, the use of SVM is more contrastive.

This paper uses the LIBSVM [29], developed by Dr. Zhiren Lin from National Taiwan University.

## 5. Experimental Design and Result Analysis

### 5.1. The Source and Composition of Sound Sample Data

The experimental data include the sound from the network and the audio clips collected in the laboratory. Audio data contain the following pests, which can change forest and urban ecosystems: iron beetle (Anthurium andraeanus), mountain skin beetle, red-necked longicorn beetle, Asian longhorn beetle and Chinese awnless beetle.

The data acquisition process from the laboratory is as follows. In order to pick up the faint vibration of trunk borers with high sensitivity, we chose the piezoelectric accelerometer YD-189-5 as the signal acquisition equipment. With the same accuracy, sensitivity and measurement, piezoelectric accelerometer YD-189-5 has the advantage of a low price. The specific parameters are as follows: sensitivity: 5.015 v/g; frequency sensing range: 0.2–5000 Hz; maximum lateral sensitivity: <5%; maximum allowable acceleration: 5 g. The temperature of signal acquisition is controlled at 26–36 °C. This is to prevent the trunk borer from going into hibernation, so that it is unable to effectively collect the signal. Then, we remove the bark from the tree and glue the sensor to the flat trunk. The collected signal is stored in the storage device after being amplified by a power amplifier. For follow-up experiments, we have prepared some audio data that do not contain the sound of trunk borer pests. Inserting the prob into a tree free of trunk borer, the recording of sound clips from trunk borers can easily be obtained from trees that are not infested by trunk borers.

We also found an interesting dataset on the Internet. In this dataset, trunk borer sound from different acoustic environments, including jungles, urban motorcycle roads, urban pedestrian alleys, and roadsides where children laugh, was collected. It records audio data from the past six months. The data of the dataset were from a record database provided by a device that periodically records and wirelessly transmits short records of the internal vibration of trees to the cloud server. Specifically, the device includes a piezoelectric sensor that records vibrations caused by pests eating trees. Vibrations were picked up by drills or metal foil acting as acoustic couplers, recorded, compressed in Ogg format, and then uploaded to the cloud server (as is shown in Figure 7), where they were decompressed, recorded and classified. The server was set up to record the time of uploading by listening equipment and the time of uploading content at the same time, and users can monitor remotely to infer the state of trees infested by withered wood insects. The device's SIM card has global coverage, so the device placed on the tree can communicate with the server at any time. The listening device has embedded solar panels, which can provide sufficient power for its own low-power electronic circuits. Therefore, it can be placed on the target tree for a long time. Some devices with GPS location function will be displayed on the server's world map.

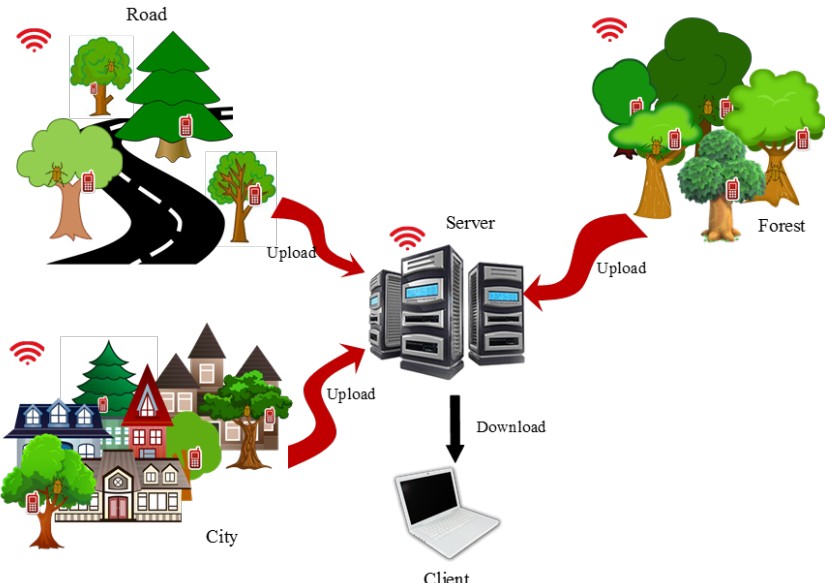

**Figure 7.** Schematic diagram of field audio signal acquisition process.

### 5.2. Optimizing Meta-Parameters of SVM Using GA

The basic idea of supporting vector machine is to find a hyperplane to maximize the classification interval in order to obtain the highest classification accuracy. For the data with nonlinear separability, the samples in the input space can be mapped to the high-dimensional feature space by nonlinear transformation. The nonlinear classification problem is transformed into the linear classification problem. This is nonlinear SVM. This kind of nonlinear transformation is usually realized by kernel function. The kernel functions of support vector machines are varied. Different kernel functions are accompanied by different nonlinear mapping methods with different hyperparameters. Common kernel functions include polynomial function, radial basis function (RBF), Sigmoid function, and so on. The mathematical expression of RBF kernel function [30] is

$$K(x_i \cdot x) = \exp\left(-\frac{\|x_i - x\|^2}{2\sigma^2}\right) \tag{17}$$

where $\sigma$ is a free parameter to indicate the variance of kernel. Therefore, for SVM using the RBF kernel function, there are two parameters to be determined, $\sigma$ and penalty factor $C$. Generally speaking, with the increase in $C$, SVM will reduce the misclassification of training data. However, it will increase the possibility of over-fitting. On the contrary, if the value of C is too small, SVM is easy to underfit, affecting the classification performance. Thus, it can be seen that the selection of meta-parameter values is very important to the performance of support vector machine. As a classical parameter optimization method, genetic algorithm (GA) is based on biological evolution [31]. Compared with other types of heuristic algorithms (such as particle swarm optimization), it has good convergence, robustness and high accuracy. This method has a wide range of applications, strong expansibility, and is easy to combine with other methods. The research shows that this method has the possibility of parallel computing. If implemented, this will greatly improve the computing speed and the robustness of the algorithm. This is also the main research topic of the GA method at present. In this study, the GA algorithm is used to select SVM meta-parameters, and the method flow is shown in the Figure 8 .

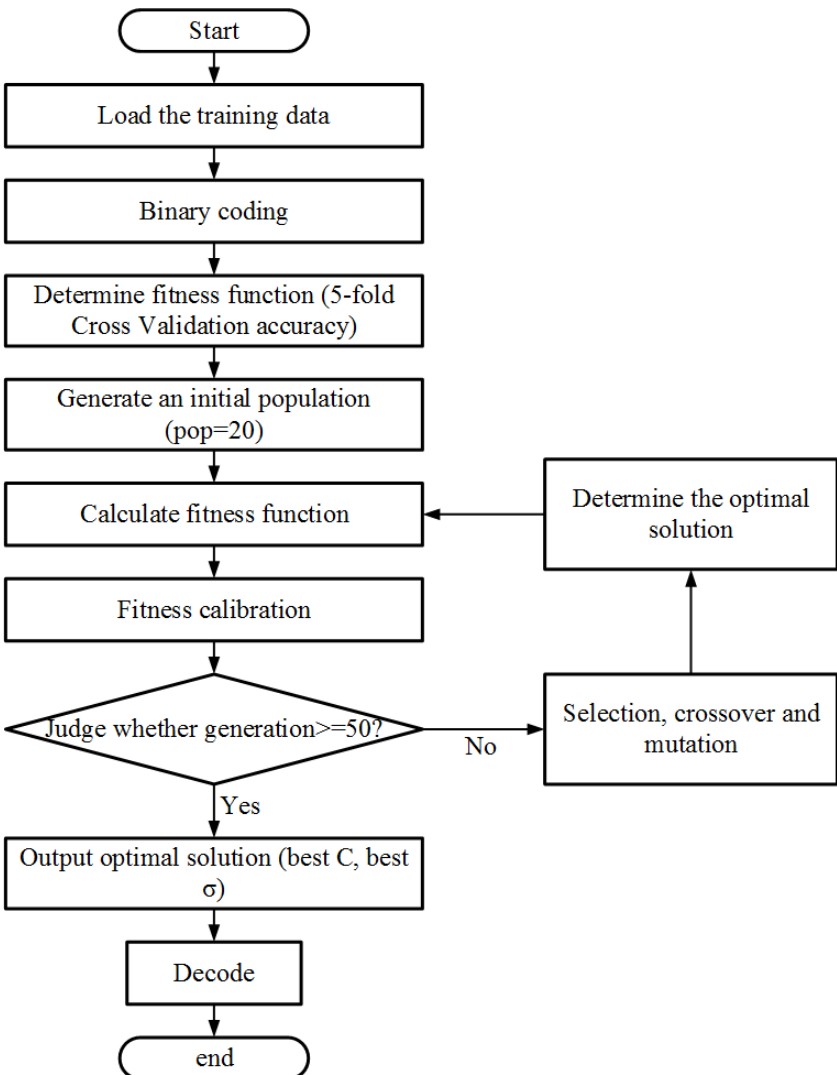

**Figure 8.** Flowchart of GA to optimize parameters *C* and *σ* of SVM.

When using the GA algorithm to select SVM meta-parameters, we should first confirm the value range of GA meta-parameters, including the maximum number of iterations, the amount of primary population, mutation probability, etc. Then, the training samples are binary coded and evenly divided into five equal parts, using the method of cross-validation to maximize the generalization ability of the training model. After repeated genetic iterations, the optimal individual of the last generation population was obtained. Finally, after the decoding operation, the optimal parameters of best *C* and best *σ* are obtained. In the optimization process, the penalty factor *C* is set at (0,200) and *σ* is set at (0,0.25). Figure 9 shows the optimization process of a set of experimental data. The results show that the SVM meta-parameter selection method based on GA can effectively improve the generalization ability of the support vector machine.

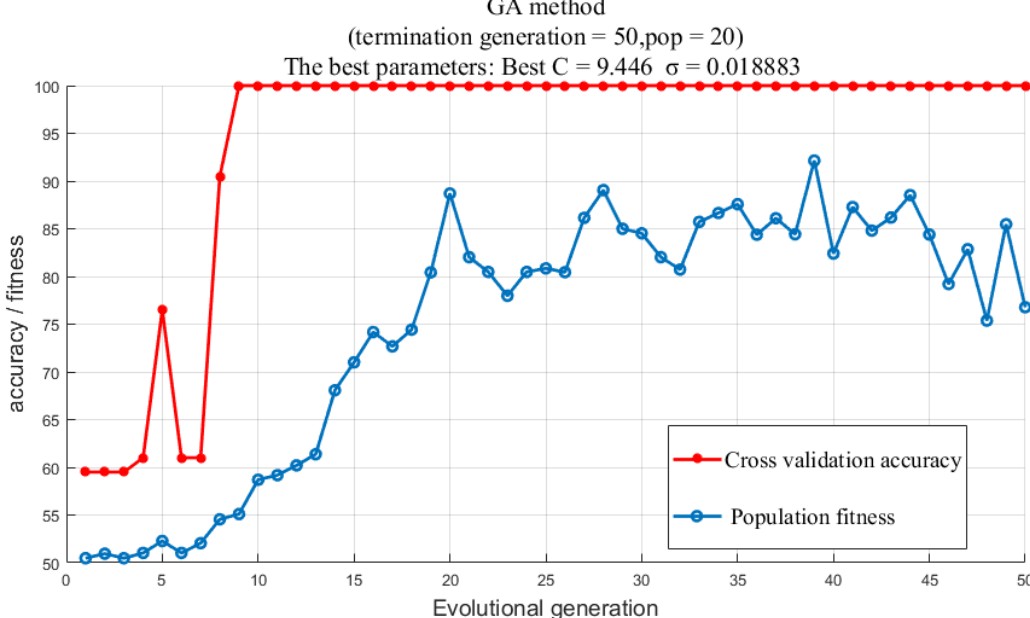

**Figure 9.** Genetic algorithm parameter optimization line chart.

### 5.3. Experiment in Laboratory

In this section, some experimental results are presented to prove the superiority of the proposed method over competitive methods in hash acoustic environments. In the first experiment, improved PNCC features, original PNCC features and MFCC features were extracted from the audio data of trunk borer pests in the laboratory environment. Then, these features were normalized and dimensionally reduced by principal component analysis (PCA) (as shown in Figure 10). Finally, SVM was used for classification. In the previous section, there is a detailed SVM parameter selection and training process. The signal processing flow of this experiment is shown in Figure 11. To show the efficiency of audio detection, we introduce the following Equation [32]

$$DAAS = \frac{NPRTS}{NATS} 100\% \tag{18}$$

where *DAAS* denotes the detection accuracy of acoustic signal, *NPRTS* denotes the number of properly recognized test samples, and *NATS* denotes the number of all test samples. Next, in order to better highlight the differences between the methods, we count the average detection accuracy of the above three methods under different signal-to-noise ratio (SNR) noise. For this purpose, the following Equation was introduced

$$ADAAS = \frac{\sum_{n=1}^{N} DAAS_n}{N} 100\% \tag{19}$$

where *ADAAS* denotes average detection accuracy of acoustic signal, $DAAS_n$ denotes detection accuracy under the interference of type *n* noise and *N* denotes the kind of noise contained in the experiment. Table 1 shows the experimental results of the detection accuracy of the three methods in different noise environments.

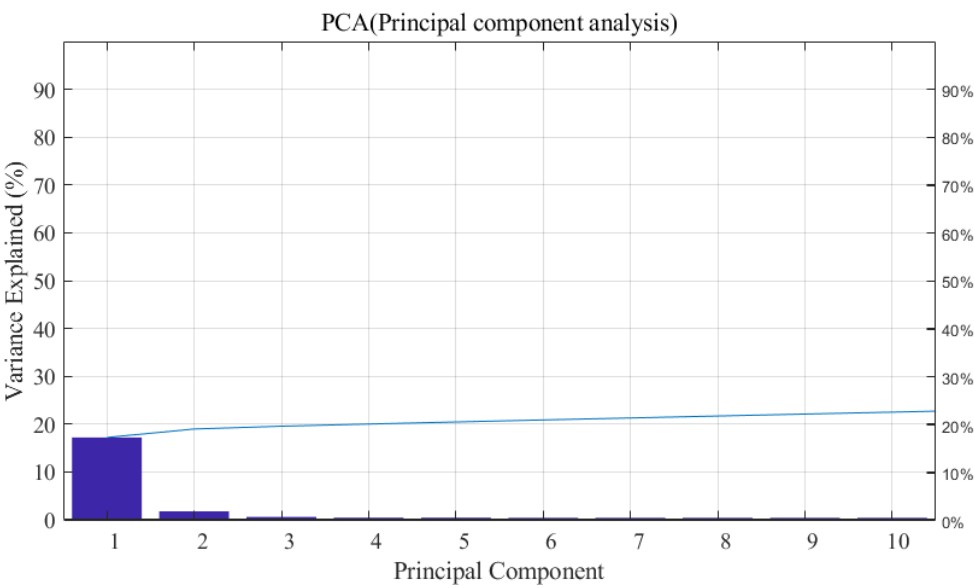

**Figure 10.** PCA principal component analysis.

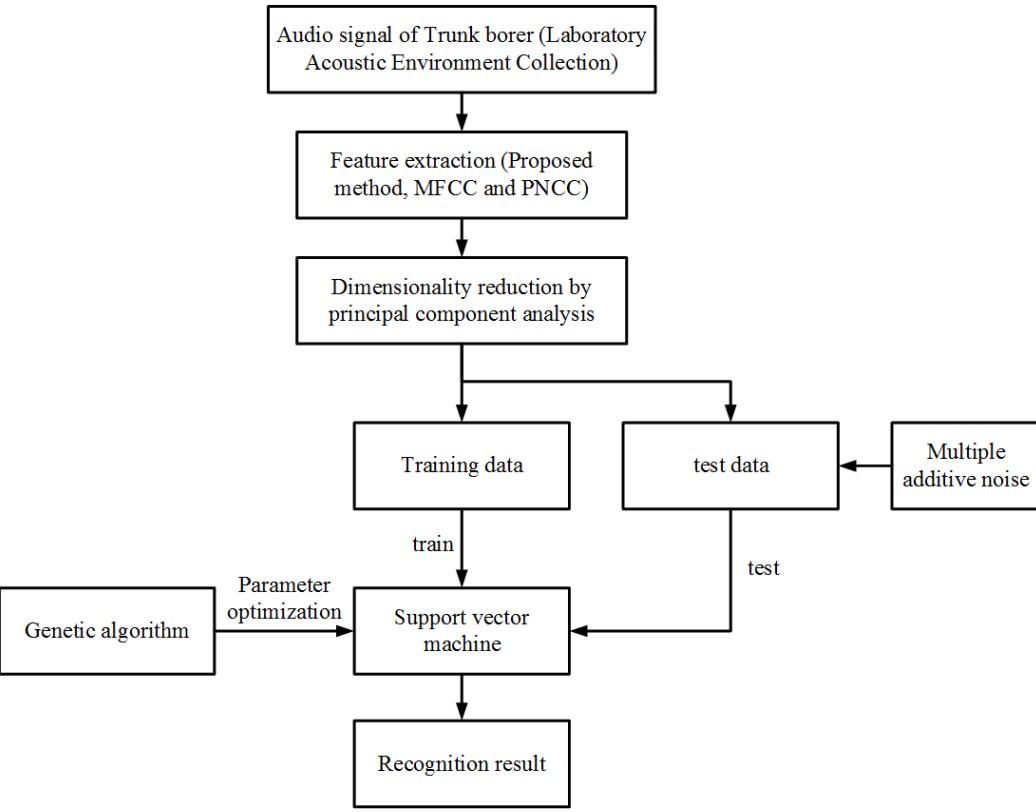

**Figure 11.** Signal processing flow chart.

Compared with the data in Table 1, the proposed method had stronger noise robustness than the other methods. The proposed method improves the detection accuracy under different reverberations, with retaining the excellent white-noise- and pink-noise-suppression ability of the original PNCC method. In practical applications, whether in the laboratory environment or in the field environment, most of the sound signal acquisition equipment will collect more or less white noise when collecting audio signals. Concretely, professional audio acquisition equipment for trunk pests collect less white noise, such as AED2010L. However, its disadvantage is also obvious; that is, the cost is too high. There-

fore, it is not suitable for large-scale audio detection. In this paper, the vibration signals collected by piezoelectric sensors often contain more white noise. However, the detection method based on improved PNCC features has strong robustness to white noise, which greatly improves the detection accuracy of audio vibration signals. At the same time, the low-cost characteristic of piezoelectric sensor makes it possible to intercept trunk borers on a large scale.

**Table 1.** Detection accuracy of acoustic signal (DAAS) under noise reverberation

| Noise /Method /SNR(DB) | | 10 | 5 | 2.5 | 0 | −2.5 | −5 | −7.5 | −10 | −15 |
|---|---|---|---|---|---|---|---|---|---|---|
| NOISEX-92 white noise | MFCC | 100% | 100% | / | 97.5 % | / | 90% | / | 50% | 50% |
| | original PNCC | 100% | 100% | / | 100% | / | 100% | / | 100% | 100% |
| | improved PNCC | 100% | 100% | / | 100% | / | 100% | / | 100% | 99.5% |
| NOISEX-92 factory noise | MFCC | 100% | 100% | / | 50 % | / | 50% | 50% | 50% | / |
| | original PNCC | 100% | 100% | / | 98% | / | 86% | 50% | 50% | / |
| | PNCC + PCA | 100% | 100% | / | 98.5% | / | 91% | 50.5% | 50% | / |
| Storm reverberation | MFCC | 100% | 100% | / | 85 % | 50.5% | 50% | / | 50% | / |
| | original PNCC | 100% | 100% | / | 82% | 64% | 50% | / | 50% | / |
| | PNCC + PCA | 100% | 99.5% | / | 95.5% | 73% | 57.5% | / | 50% | / |
| Heavy rain reverberation | MFCC | 100% | 100% | 98.5 % | 50% | / | 47.5% | / | 50% | / |
| | original PNCC | 100% | 100% | 100% | 90.5% | / | 78% | / | 62% | / |
| | PNCC + PCA | 100% | 100% | 100% | 99% | / | 85% | / | 66.5% | / |
| Thunder reverberation | MFCC | 100% | 99% | / | 97.5 % | 50% | 51.5% | / | 50% | / |
| | original PNCC | 99.5% | 100% | / | 97% | 78% | 62% | / | 53% | / |
| | PNCC + PCA | 100% | 99.5% | / | 97% | 95% | 82% | / | 50.5% | / |
| Stream reverberation | MFCC | 100% | 100% | / | 100 % | / | 98.5% | / | 50% | 50% |
| | original PNCC | 100% | 100% | / | 100% | / | 100% | / | 99% | 100% |
| | PNCC + PCA | 100% | 100% | / | 100% | / | 100% | / | 99% | 100% |
| NOISEX-92 pink noise | MFCC | 100% | 100% | / | 52.5 % | / | 50% | / | 50% | 50% |
| | original PNCC | 100% | 100% | / | 100% | / | 100% | / | 100% | 50% |
| | PNCC + PCA | 100% | 100% | / | 100% | / | 100% | / | 100% | 50% |
| Train reverberation | MFCC | 100% | 97.5% | / | 60 % | 50% | 50% | / | 50% | / |
| | original PNCC | 97.5% | 97.5% | / | 82% | 50% | 50% | / | 50% | / |
| | PNCC + PCA | 99.5% | 97% | / | 91% | 52.5% | 50% | / | 50% | / |
| TDAAS | MFCC | 100% | 99.5625% | / | 74.0625% | / | 60.9375% | / | 50% | / |
| | original PNCC | 99.625% | 99.6875% | / | 93.6875% | / | 78.25% | / | 70.5% | / |
| | PNCC + PCA | 99.9375% | 99.5% | / | 97.625% | / | 83.1875% | / | 70.75% | / |

*5.4. Experiment in Field*

The experimental results show that the proposed method has significant advantages in the acoustic environment of artificial noise and reverberation. The original data of the above experiments come from a good laboratory acoustic environment. In order to verify the effect of the new method in hash outdoor acoustic environment, we designed another experiment. Here, the harsh acoustic environment refers to the recording environment with a variety of additive, multiplicative noise, reverberation and other interference. We deliberately chose audio recorded from different outdoor locations as the training and test data of the classification model. Among them, the audio collection environment of training data included noisy streets in the center of the city, streets in the suburbs of the city, bad weather conditions with wind and rain, children playing nearby, trees in the forest, etc. The audio collection environment of the test dataset was near the alleys in the city.

According to Equation (18), the experimental results showed that the classification accuracy using the improved PNCC feature was 88% on the test set. The traditional PNCC method obtained 78% accuracy in tests. However, the accuracy of using the MFCC feature in the training set was 50%. It was found that the classification labels given by the model were designated as "the presence of trunk borer", and this feature cannot be used in the training of the classification model. It was revealed that, under the condition of a harsh outdoor acoustic environment, MFCC features cannot be used to identify trunk borers.

## 6. Conclusions

One of the basic aims of this study is to analyze and process the audio signals of several typical trunk borers so as to achieve the purpose of detecting trunk borers. However, the process of signal acquisition is greatly affected by background environmental factors. In practical application, sound monitoring of trunk borers is often not guaranteed to be carried out in an interference-free environment. In order to solve this problem, work has been carried out to explore the feasibility of applying these advanced technologies in related fields, such as acoustic detection and sound classification, which are less affected by environmental factors, to this study. Finally, we found that the PNCC method is generally applicable to this study. In view of the fact that the design of this method is close to the characteristics of human speech, we think that it should be improved, in order to have a better ability to detect dry borer pests. Therefore, the work of improving the PNCC method to make it suitable for audio detection of trunk borer is carried out. In this work, firstly, the signal of trunk borers is briefly analyzed and, combined with the signal properties, an improved method based on the dmey wavelet system is established. The complete detection method includes two parts: feature extraction and classification. The performance of the classifier determines the detection accuracy. How to select and improve the performance of the classifier is another main work of this research. In this part, we compare the advantages and disadvantages of the commonly used methods to determine SVM as the classifier, and use GA to optimize the SVM meta-parameters to obtain the best classification results. Finally, in order to verify the effectiveness of the proposed method, we carried out experiments. The experimental results show that the proposed method is always more effective than the traditional sound detection technique for trunk borer pests. This provides an idea regarding the detection of trunk borers in a bad acoustic environment. In the process of research, we find that the proposed improved PNCC method inherits the strong anti-interference ability of the traditional PNCC method to white noise. This feature can significantly reduce the cost of audio recording equipment for trunk borer pests. In fact, cheap audio recording equipment, combined with the improved PNCC method, can not only detect whether the target tree contains trunk borers, but can also be used to monitor trunk borers on a large scale. It should be noted that the method proposed in this study has only been proven to be effective for the four trunk pests mentioned above. There is no extensive verification of whether it is applicable to other research objects. This is also a problem that needs to be discussed in future work. In future research work, we will further improve the existing methods in the following directions. First of all, at the data level, more species of trunk borer will be considered to verify the effectiveness of the method. In addition, the current research on acoustic detection of trunk borers is mainly based on a single signal source. However, multi-source signal often provide more information than a one-source signal. Therefore, the detection method based on multi-source signals will be developed. Then, at the feature level, we mention the idea of improving PNCC based on more advanced signal processing methods, which is an important part of the following work. Finally, at the classifier level, the method should distinguish the species and behavior of trunk borers in adverse acoustic environments.

**Author Contributions:** Methodology, H.Z. (Huanyu Zhou); validation, H.Z. (Huanyu Zhou), Z.H.; formal analysis, H.Z. (Huanyu Zhou); resources, H.Z. (Huanyu Zhou) and X.L.; data curation, H.Z. (Huanyu Zhou) and X.L.; writing—original draft preparation, H.Z. (Huanyu Zhou); writing—review and editing, H.Z. (Huanyu Zhou) and Z.H.; supervision, L.S. and D.Z.; project administration, L.S.; funding acquisition, H.Z. (Hongwei Zhou). All authors have read and agreed to the published version of the manuscript.

**Funding:** This research was funded by "Heilongjiang Provincial Natural Science Foundation of China", grant number: YQ2020C018, and by "The Fundamental Research Funds for the Central Universities", grant number: 2572019BF08.

**Conflicts of Interest:** The authors declare no conflict of interest.

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
