# Peer review of "Improved Power Normalized Cepstrum Coefficient Based on Wavelet Packet Decomposition for Trunk Borer Detection in Harsh Acoustic Environment"

_applsci, doi:10.3390/app11052236_

Round 1

Reviewer 1 Report

This manuscript proposed a novel improved power normalized cepstrum coefficient based on wavelet packet decomposition for trunk borer detection. In addition, the support vector machine was employed to implement the classification task. To improve the generalisation capacity of SVM model, the genetic algorithm was used to optimize the hyperparameters of SVM. Finally, the experimental study was conducted to validate the performance of proposed method with satisfactory results. Overall, the topic of this research is interesting, and the structure of manuscript was well organized. My detailed comments are given as follows.

  1. Please illustrate the main innovation of this study. Why were wavelet packet transform and SVM selected for the task of interest? Why not other similar methods?
  2. The following reference is suggested to be included to better illustrate the wavelet packet transform.

https://doi.org/10.1061/(ASCE)AS.1943-5525.0001019

  1. Please explain how to define the decomposition level in this study.
  2. Add more contents about how the hyperpareters of SVM can affect the prediction performance. Follow the paper Multi-Image-Feature-Based Hierarchical Concrete Crack Identification Framework Using Optimized SVM Multi-Classifiers and D–S Fusion Algorithm for Bridge Structures to see the detailed explanation.
  3. In this study, the GA was selected to optimize the SVM. Please explain why GA is considered and its superiority over other similar optimization algorithms.
  4. More future research should be included in conclusion part.

Reviewer 2 Report

-please add block diagram of the proposed research step by step ;;; what is the result of paper?;;;
-please add block diagram of the proposed method;;;
-please add photo/photos of application of the proposed research ;;;;
-please add sentences about future analysis;;;
-references should be 2018-2021 Web of Science about 50% or more ;;
-Please compare with other methods, justify. Advantages or Disadvantages;;;
for example:

1) Acoustic fault analysis of three commutator motors, Mechanical Systems and Signal Processing, vol. 133 art. no. 106226, 2019,
https://doi.org/10.1016/j.ymssp.2019.07.007

-Conclusion: point out what are you done;;;;
-Could you use the proposed method for other acoustic signals?

Round 2

Reviewer 1 Report

The authors well addressed the reviewer's comment. I suggest this revised version can be accepted for publication in Applied Sciences.

Reviewer 2 Report

-it is good idea to add photo of analysed objects

-it is good idea to compare with other methods based on acoustic analysis, for example

Recognition of Acoustic Signals of Loaded Synchronous Motor Using FFT, MSAF-5 and LSVM, Archives of Acoustics, 2015, 40 (2), pp. 197-203.
https://doi.org/10.1515/aoa-2015-0022
